# Use of GnRH Treatment Based on Pregnancy-Associated Glyco-Proteins (PAGs) Levels as a Strategy for the Maintenance of Pregnancy in Buffalo Cows: A Field Study

**DOI:** 10.3390/ani12202822

**Published:** 2022-10-18

**Authors:** Corrado Pacelli, Vittoria Lucia Barile, Emilio Sabia, Anna Beatrice Casano, Ada Braghieri, Valeria Martina, Olimpia Barbato

**Affiliations:** 1School of Agricultural, Forest, Food, and Environmental Sciences, University of Basilicata, 85100 Potenza, Italy; 2Consiglio per la Ricerca in Agricoltura e l’Analisi dell’Economia Agraria (CREA)—Research Centre for Animal Production and Aquaculture, 00015 Rome, Italy; 3Department of Veterinary Medicine, University of Perugia, 06126 Perugia, Italy

**Keywords:** buffalo, pregnancy, PAGs, embryonic mortality, GnRH

## Abstract

**Simple Summary:**

Embryonic mortality (EM) is one of the leading causes of infertility in modern breeding. This results in a reduction in reproductive performance, significant economic losses and can affect the environmental impact. The introduction of intensive breeding systems and the selection of high-production animals have accentuated EM, also in buffalo. This study aimed to verify if GnRH treatment at 35 days after artificial insemination of subjects identified at risk of embryonic mortality on the basis of pregnancy-associated glycoprotein (PAG) levels, could reduce EM in buffalo.

**Abstract:**

The aim of the study was to evaluate the effects of GnRH administrated at day 35 after artificial insemination (AI) on the reproductive performance of buffalo cows. In ten buffalo farms in the period January–February, 481 buffalo cows were subjected to estrus synchronization protocol and fixed-time artificial insemination (Ovsynch–TAI program). Radioimmunoassays (RIA) for pregnancy-associated glycoproteins (PAGs) were used to detect pregnancy at day 28 after AI (cut-off value: ≥1 ng/mL). Among pregnant subjects, those with PAG values between 1 and 2.5 ng/mL were considered at risk of embryonic mortality (EM) and were assigned into two groups: treated (T; *n* = 57) control (C; *n* = 57). Treated buffaloes received 0.01 mg of buserelin acetate intramuscularly on day 35 after AI, whereas control buffaloes received no treatment. The pregnancy diagnosis was confirmed at day 60 through PAGs level and rectal palpation. The treatment with GnRH had a significant effect (*p* < 0.001) in reducing EM. Between days 28 and 60 after AI, the animals that experienced EM were only 2/57 in the T group, while were 13/57 in the C group. Moreover, GnRH treatment produced a significant increase (*p* < 0.001) in the PAG concentration between day 28 and day 60. Administration of GnRH at day 35 after AI in animals considered at risk of low embryo survival based on PAG levels allowed a reduction in pregnancy losses and improved the pregnancy rate during low-breeding season in buffalo.

## 1. Introduction

The efficiency of reproductive management is the main determining factor of profitability in an intensive dairy herd. Oestrus detection and fertilization success are important pre-requisites to obtain good pregnancy rates. Once fertilization occurs, the next impediment is embryo loss. There are three recognized critical stages in the early embryo development whereby loss can occur: around days 5–7, when the morula is developing into blastocysts; around days 15–16, when the embryo must prevent the prostaglandin (PG) F2α secretion from the uterus, avoiding the corpus luteum (CL) regression and allowing the establishment of pregnancy; during the 25–40-day period, whereby failures in the maternal-embryo connection can occur [1]. Embryonic mortality (EM) following day 28 of pregnancy has a greater economic impact on dairy herd management, because the delay in resynchronization and rebreeding of animals prolongs their intercalving. The identification of EM anticipates a new insemination of non-pregnant animals and reduces the environmental impact [2,3].

The introduction of intensive breeding systems and the selection of high-production animals have accentuated EM, either in bovine or in buffalo, since high-production animals are much more at risk of EM than medium- or low-production animals [4,5,6,7]. Moreover, the occurrence of EM is higher when the animals are not mated during their natural reproductive period. This is the case of buffalo, whereby their reproductive efficiency changes throughout the year, showing a lower efficiency during the spring–summer season (low-breeding season) with the daylight lengthening period [8]. In fact, in buffalo, the occurrence of EM has been reported to be between 7% and 10% during the decreasing daylight length period, corresponding to the natural breeding season [9,10], while can reach 40% during the so-called “transition period” from the breeding to the non-breeding season [11,12]. Different factors are involved in the higher incidence of EM in buffalo during the low-breeding period: a reduction in hypothalamic–pituitary axis responsiveness occurring in the daylight-lengthening period [8]; higher levels of prolactin, which is known to exert suppressive effects on the secretion of gonadotropins and ovarian steroids, thus, contributing to the summer anestrus [13]; impaired luteal function [14,15]; reduction in oocyte competence and sperm quality [16]. Human–animal interactions may also have an effect on embryonic mortality due to stress [17,18,19]. 

A key factor for the establishment of pregnancy is the maternal recognition of the presence of conceptus. In ruminants, this process requires the production of interferon-tau (IFN-τ) by the conceptus, which plays antiluteolityc effects that are essential for embryonic growth and development [20,21]. In addition to IFN-τ, the conceptus secretes several molecules during early pregnancy, among which the pregnancy-associated glycoproteins (PAGs) that are expressed in ruminants at the conceptus-maternal interface and secreted by trophectoderm binucleate cells, starting around the time of beginning of implantation and early accumulated in maternal blood [22]. The PAGs play an important role in the function of the placenta in ruminants. For this reason, PAGs have been studied extensively for their application in pregnancy diagnosis in ruminant species including cattle, sheep, goats, buffalo, bison, moose and elk [22,23].

Distinct PAG molecules are also purified and characterized from water buffalo placenta (wbPAG), allowing the development of different RIA systems [24,25]. Purified and semi-purified preparations have been used to immunize rabbits in order to obtain antisera (AS), which has led to the development of homologous and heterologous RIA. The first RIA system adopted for detecting PAG molecules in buffalo species was RIA-706, which uses antisera raised against caprine PAGs (caPAG55kDaþ62kDa) and purified bovine PAG as a tracer. More recently, the isolation and purification of PAGs from buffalo placenta allowed for the development of a specific RIA system for buffalo. Three polyclonal antisera (AS#858, AS#859 and AS#860) were obtained against distinct buffalo PAG fractions (wbPAG76kDa_D, wbPAG65kDa_E and wbPAG58kDa). The highest dilution of primary antiserum (1:840,000) was obtained with AS#860, allowing distinguishing quantitative differences in buffalo PAG concentrations [24,25]. Using this system, buffalo plasma PAGs profiles were described during pregnancy and in the postpartum period [26]. The PAG assay has been demonstrated to be a reliable biomarker for early pregnancy detection in buffalo [27,28,29]. Recently, Barile et al. [30] investigated the best strategy to diagnose pregnancy failures in buffalo comparing ultrasonography, progesterone (P4) and PAGs. The results showed that as a predictor for pregnancy loss, PAGs and ultrasonography are more reliable compared to P4. An ultrasonography disadvantage is that the pregnancy status is only guaranteed at the time of diagnosis. 

PAGs allowed for distinguishing between buffalo that experienced embryonic mortality and those that carried on pregnancy starting from 25 days of gestation, defining the optimal cut-off value for predicting embryo loss in 1,1 ng/mL and 2.2 ng/mL at day 25 and 28, respectively, post-artificial insemination (AI) [30]. PAGs indicate embryo wellbeing [31,32], and therefore, the decrease in its plasma concentrations is a prognostic sign of pregnancy failure.

The gonadotropin-releasing hormone (GnRH) is a deca-peptide produced by the hypothalamus and released to the pituitary gland through the portal circulatory system. This releasing hormone stimulates the release of the luteinizing hormone (LH) and folliclestimulating hormone (FSH) that are fundamental for the regulation of ovarian function. For this reson, the GnRH is utilized in the field to apply breeding programs as well as to enhance embryo survival rates after AI [33].

Several reports have described the use of a single GnRH injection between days 11 and 14 after AI to increase pregnancy rates in cattle [33,34,35,36]. In buffalo, Campanile et al. [37] showed that treatment with a GnRH on day 5 after AI increased P4 secretion and the chance of pregnancy in buffaloes that mated during the period of increasing daylight length. The same authors reported that the delayed treatment with GnRH on day 25 after AI resulted in a reduction in EM in buffalo [38].

The scientific basis for GnRH treatment is to enhance embryo survival rates by delaying the luteolytic mechanism [39] that could occur due to failure in the maternal recognition of pregnancy.

Considering that the reduction in EM is an important factor in the management of reproductive efficiency of a herd and that PAGs could be utilized as a predictor for pregnancy loss, with this investigation, we wanted to verify if a treatment with GnRH in buffaloes that show low PAGs values at day 28 after AI could reduce embryonic losses. 

## 2. Materials and Methods

The animals utilized in this research were supervised in compliance with Italian laws and regulations concerning experimental animals (D.Lgs. 26/2014). The experimental design was performed according to good veterinary practices under farm conditions. The University of Basilicata (UNIBAS) Committee of Ethics in Animal Research assessed and approved the experimental procedures (Protocol code: OPBA 02_2022_UNIBAS).

### 2.1. Animals and Experimental Design

The trial was carried out in ten buffalo farms located in Southern Italy, from January to February—the months that correspond to the transition from the breeding to non-breeding season for buffalo in Italy. The breeding system did not differ between farms. Buffaloes belonging to the Italian Mediterranean breed were kept on a loose-housing system, fed ad libitum one a day on a total mixed ration (0.90 UFL and 15% crude protein/Kg dry matter) and milked twice a day. All lactating buffaloes were enrolled to be subjected to synchronization and the fixed-time AI program. Before estrous synchronization, buffaloes underwent a routine examination to exclude any reproductive disorders.

Four hundred and eighty-one animals were considered suitable for the AI program. Buffaloes were synchronized with the Ovsynch protocol, as reported by Neglia et al. [15]. Briefly, animals received i.m. GnRH agonist (buserelin acetate, 10 μg; Ovulike, T.P. Whelehan Son & Co., Ltd., Bracetown Business Park, Clonee, Co., Meath, Ireland) on day 0, Cloprostenolo (Estrumate, 500 μg; MSD Animal Health, Segrate, Milano, Italy) on day 7, and GnRH agonist (10 μg) again on day 9. Animals were artificially inseminated using frozen-thawed semen approximately 16 h after the administration of the second GnRH.

On days 28 after AI, blood samples were collected from the coccygeal vein in 10 mL BD vacutainer serum tubes and immediately sent to the laboratory of the Department of Veterinary Medicine, University of Perugia, for PAG determination. 

Among the animals diagnosed as pregnant by the PAG assay, as described in the following 2.2 paragraph, those considered at risk of EM were divided in two groups, control (C, *n* = 57) and treated (T, *n* = 57), homogeneous in terms of the number, distance from calving and parity. In the T group, animals received i.m. 10 μg of buserelin acetate (Ovulike, T.P. Whelehan Son & Co., Ltd., Bracetown Business Park, Clonee, Co., Meath Ireland) on day 35 after AI, whereas in the C group, animals received no treatment. On day 60 after AI, all subjects of the two groups (T and C) had a new blood sample collection, as described above, to assess PAG concentration for the follow-up of pregnancy. 

All buffaloes subjected to the AI program were checked by rectal palpation at day 60 post-AI to confirm pregnancy status. 

### 2.2. PAGs Radioimmunoassay

PAG concentrations were determined by RIA-706, as previously described by Perenyi et al. [40] and Barbato et al. [29]. Pure boPAG67kDa preparation was used as the standard and tracer. Iodination (Na-I125, Amersham Pharmacia Biotech, Uppsala, Sweden) was carried out according to the chloramine-T method, as previously described by Greenwood et al. [41]. The samples were assayed in a preincubated system in which the standard curve ranged from 0.2 to 25 ng/mL.

The minimum detection limit (MDL), calculated as the mean concentration minus twice the standard deviation (mean–2 SD) of 20 duplicates of the zero (B0) standard [42], was 0.3 ng/mL. The intra- and inter-assay coefficients were 2.7% and 6.9%, respectively.

Based on the cut-off value of ≥1 ng/mL, buffaloes were considered non-pregnant when concentrations remained very close to zero, and pregnant when concentrations were higher than 1 ng/mL at days 28 and 60. When PAG concentrations were ≥1 ng/mL at day 28 and dropped under 0.2 ng/mL by day 60, EM was considered to have occurred.

Based on a previous study [30], which identified a PAG value of 2.2 ng/mL as predictive of EM, we have defined a PAG concentration range of 1.0–2.5 ng/mL as a risk factor for EM. 

### 2.3. Statistical Analyses

Data were analyzed using a Statistical Analysis System (SAS Institute, Cary, NC, USA; SAS\STAT 2000).

The conception rate was analyzed using the X2 test. PAG concentrations, reported as the mean ± standard error of the mean, were analyzed by ANOVA using the general linear model procedure. The blood values of PAGs regarding the effects of farms, calving order and distance from calving were processed according to ANOVA analysis of the variance model with interactions.

## 3. Results

Based on the PAG concentration recorded at day 28 after AI, the number of buffaloes diagnosed as pregnant was 276, with a total conception rate (CR) of 57.4% (Table 1). Neither the farm, nor the distance from calving, nor the parity affected CR. At day 60 after AI, buffalo cows confirmed pregnant by rectal palpation were 258 (53.6% CR). Excluding the buffaloes that were subjected to GnRH treatment, in order to avoid the possible effect of the treatment on pregnancy maintenance, the CR at day 60 was 47.9%. Therefore, without any treatment, 7.3% of EM was recorded (Table 1).

Out of the 276 buffalo cows diagnosed as pregnant at day 28 by PAG measurement, 162 had a PAG concentration of >2.5 ng/mL, while 114 animals had PAG concentrations between 1.0 and 2.5 ng/mL, and thus, were considered at risk of low embryo survival and were assigned to the experimental groups (T and C). Out of the 162 buffaloes that had PAG concentrations of >2.5 ng/mL at day 28, 3 experienced EM while the others maintained their pregnancy, as confirmed by rectal palpation. The treatment with GnRH (Table 2) had a significant effect (*p* < 0.001) in reducing EM. In fact, between days 28 and 60 after AI, the animals that experienced EM were only 2/57 in the T group, while were 13/57 in the C group. Moreover, GnRH treatment (T group) produced a significant increase (*p* < 0.001) in the PAG concentrations between day 28 and day 60, as reported in Table 3.

## 4. Discussion

To the best of our knowledge, this is the first study on the effect of GnRH treatment in buffalo considered at risk of low embryo survival based on PAG levels.

The CR obtained at day 60 after AI (47.9%) does not differ from those reported by others authors in the same period of the year in Italy that ranged between 41.8% [43] and 48.7% [44]. The percentage of embryo mortality in our study, excluding the treated group, was 7.3%, in line with the average values reported for buffalo in previous studies conducted in the same period of year [9,10,45].

GnRH treatment seems to have a positive effect on PAG levels. The treated buffaloes showed an increase in glycoprotein concentrations between days 28 and 60 of pregnancy compared to the controls. The period of gestation during which the animals were subjected to the treatment corresponds to the phase in which the embryo moves from the histotrophic nutrition to hemotrophic nutrition, which is the exchange of nutrients between the maternal and fetal circulations within the placenta. In fact, in cow, placentation begins at 28–32 days and ends between 40 and 45 days [46,47]. The increase in PAG levels reflects the growth of trophoblastic tissue, and its concentration is related to placental development [48,49]. The trophoblastic cells, particularly the binucleate cells, have a direct role in prostaglandin (PG) metabolism being able to convert PGF2α into PGE2—which has a luteotrophic function—ensuring the CL lifespan prolongation and adequate progesterone concentrations required for the survival of the conceptus [50,51].

The study in bovine and ovine luteal cells in vitro showed that PAGs may indirectly regulate luteal progesterone production by the stimulation of PGE2 synthesis [52,53]. Austin et al. [54] also asserted the indirect luteotropic effect of PAGs in the bovine, since these glycoproteins stimulate the release of granulocyte chemotactic protein-2 (GCP-2) in endometrium, whose synthesis is induced by IFN-τ during early pregnancy. 

Previous works reported the use of GnRH to prevent EM in cattle. Early administration in the first weeks of gestation, 5th or 15th day [34] and 12th day [36], did not show any significant improvement in the pregnancy rate. The administration of GnRH by subcutaneous implant at day 27 after AI resulted instead in a significant recovery of pregnancy loss between 45 and 90 days in animals that developed accessory corpora lutea [35].

The establishment of pregnancy requires a progressive increase in progesterone secretion during the first 2–3 weeks after mating. In buffalo, different treatments prolonging the CL lifespan, and thus, supporting pregnancy, have been utilized [38,55,56]. Campanile et al. [38], reported that the GnRH treatment on day 5 after AI increases progesterone secretion and the chance of successful pregnancy in buffaloes that mate during periods of increasing daylight length (outside the breeding season) when progesterone secretion is relatively low and the incidence of embryonic mortality tends to be high. The same researchers, applying a delayed treatment on day 25 after AI, had an increased pregnancy rate [57]. In Pakistan, Arshad et al. [56] also found that the administration of GnRH at day 23 as a resynchronization strategy in buffalo enhanced the pregnancy rate on day 30 and reduced the cumulative embryonic/fetal losses on days 45, 60 or 90 post the first AI when compared with control buffaloes. 

Our study confirmed the finding that a delayed treatment of GnRH in buffalo reduces EM, but unlike others, the novelty was that the treatment was performed only on animals with low levels of PAGs. These glycoproteins reflect embryo wellbeing and, therefore, the reduction in its circulating concentrations is a sign of pregnancy failure, as previously reported in buffalo [30,58] and in bovine species [59,60].

The GnRH could induce ovulation or follicle atresia, thus, reducing the circulating concentrations of follicular estrogens, leading to a reduction in the endometrial production of PGF2α, which is responsible for luteolysis. As estrogens fail, the uterine expression of the oxytocin receptors lacks, and consequently, its effect on PGF2α secretion lacks, ensuring CL lifespan prolongation and adequate progesterone productions for embryo survival. Thus, the exogenous GnRH administered at day 35 after AI in subjects identified as at risk of EM (PAGs level range of 1.0–2.5 ng/mL) could support pregnancy maintenance.

## 5. Conclusions

The PAG assay 28 days after AI has proved to be an effective and practical method in buffalo for predicting embryonic mortality. Its utilization as a diagnostic tool can influence management decisions in order to improve farm reproductive performance, thus, allowing the recovery of those animals that would experience pregnancy loss.

The identification of subjects at risk of EM allows the pharmacologically intervention only on the identified subjects instead of on the whole herd, reducing the worthless use of drugs and respecting the economic and environmental sustainability of the breeding system. A single injection of GnRH at day 35 after AI resulted in an increase in PAG concentration and recovery of embryo wellbeing, allowing a reduction in pregnancy loss in early pregnancy and improving the pregnancy rate in low-breeding seasons in buffalo.

In conclusion, the luteotropic effect of exogenous GnRH may add to the luteotropic effect of PAGs in supporting pregnancy maintenance in animals considered at risk of low embryo survival.

## Figures and Tables

**Table 1 animals-12-02822-t001:** Reproductive performance on all buffalo cows (total) and on buffalo without any treatment (*w/o* T group) that underwent artificial insemination (AI).

Parameters	Total	*w/o* T Group
AI (n)	481	424
Pregnant at day 28 ^a^ (n)	276	219
Conception rate at day 28 (%)	57.4	51.6
Pregnancy at day 60 ^b^ (n)	258	203
Conception rate at day 60 (%)	53.6	47.9
Embryonic/fetal mortality (%)	6.5	7.3

^a^ based on PAGs, ^b^ based on rectal palpation.

**Table 2 animals-12-02822-t002:** Effect of treatment with GnRH (T) at day 35 after AI in pregnant buffalo cows at risk of low embryo survival.

Parameters	Treated Group (T) N (%)	Control Group (C) N (%)
Buffalo cows	57	57
Pregnancy rate at day 60	55 (96.5) *	44 (77.2) *
Embryonic/fetal mortality	2 (3.5) *	13 (22.8) *

* *p* < 0.001 in row.

**Table 3 animals-12-02822-t003:** Change in PAG concentrations (ng/mL) (mean ± SE) in GnRH-treated (T) and in the control (C) of pregnant buffalo cows at risk of low embryo survival.

Group	Day 28 ng/mL	Day 60 ng/mL	Increase
Treated (T) (*n* = 57)	1.45 ±0.57	15.21 ± 0.97 *	13.75 ± 0.95 *
Control (C) (*n* = 57)	1.44 ± 0.57	8.74 ± 0.97 *	7.30 ± 0.95 *

* *p* < 0.001 in column.

## Data Availability

Not applicable.

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
