# Peer review of "Use of GnRH Treatment Based on Pregnancy-Associated Glyco-Proteins (PAGs) Levels as a Strategy for the Maintenance of Pregnancy in Buffalo Cows: A Field Study"

_animals, 2022, doi:10.3390/ani12202822_

Round 1

Reviewer 1 Report

The authors report the effect of delayed GnRH in buffalo with a high embryo loss risk based on PAGs levels. In my opinion the manuscript brings new interesting information of buffalo reproduction and for this reason I suggest to accept it after minor revisions.

Specific comments

Lines 124-134: Please explain, why you preferred  to use onlòy a predictive diagnosis by PAGs dosage and did not confirm it with ultrasound diagnosis?

Lines 137-141: Please remove this sentence and insert it before the Animal and expreimental..... paragraph

Lines 166-168: The statistical analisys performed in this manuscript did not permit to assert "Neither the farm, nor the distance .........". It would be advisable to perform an ANOVA analysis with interaction at least for the PAGs levels, method used to establish potential pregnancy.

Lines 173-176: Please reformulate the sentence becuase the potential pregnancy diagnosi were performed by PAGs dosage.

Reviewer 2 Report

This is an interesting study to reduce embryo mortality in buffalo. Administration of GnRH at day 35 post-AI in animals considered at risk of low embryo survival based on PAGs levels, allowed a significant reduction of pregnancy losses on day 60 post-AI. The manuscript is well written and presented. The acceptance of the manuscript in the present form can be greatly supported.

Minor point: Line 221- corpora lutea

Reviewer 3 Report

Minor observations

Are you sure that the abreviation w/o T Group (Table 1) will be clear for all the readers ?

Line 228 : I suggest to remove « rate » from the sentence

Major observation

Lines 239-243: The authors seem to suggest that GnRH can induce ovulation of the dominant follicle in pregnant buffalos. If follicular atresia might explain the GnRH effect on EM reduction, the propability of ovulation after 28 days of pregnancy might be very low in my opinion. The authors might indicate that these are hypothesis to explain GnRH effect on EM.  They can't verify their affirmation in the study design.
